# Starch and Cellulose Degradation in the Rumen and Applications of Metagenomics on Ruminal Microorganisms

**DOI:** 10.3390/ani12213020

**Published:** 2022-11-03

**Authors:** Dengke Hua, Wouter H. Hendriks, Benhai Xiong, Wilbert F. Pellikaan

**Affiliations:** 1State Key Laboratory of Animal Nutrition, Institute of Animal Sciences, Chinese Academy of Agricultural Sciences, Beijing 100193, China; 2Animal Nutrition Group, Department of Animal Sciences, Wageningen University & Research, 6708 PB Wageningen, The Netherlands

**Keywords:** rumen, starch, cellulose, microbe, enzyme, metagenomics

## Abstract

**Simple Summary:**

Starch and cellulose are the principal components in diets for dairy cows worldwide, providing the primary energy to the rumen microorganisms as well as the host. Starch and cellulose degradation in the rumen have always been of key importance for dairy cows to obtain high production performance. To improve the starch- and cellulose-degrading activities in the rumen, the amylolytic and cellulolytic microbes and the related enzymes need to be well understood. As the rapid development of sequencing technologies, bioinformatic tools and reference databases, the rumen metagenomics have made great progress in mining the rumen microbial community for novel enzymes, such as the carbohydrate active enzymes (CAZymes). This review will summarize the ruminal microbes and enzymes involved in starch and cellulose degradation. Recent studies with metagenomics techniques on CAZymes related to starch and cellulose degradation will be discussed.

**Abstract:**

Carbohydrates (e.g., starch and cellulose) are the main energy source in the diets of dairy cows. The ruminal digestion of starch and cellulose is achieved by microorganisms and digestive enzymes. In order to improve their digestibility, the microbes and enzymes involved in starch and cellulose degradation should be identified and their role(s) and activity known. As existing and new analytical techniques are continuously being developed, our knowledge of the amylolytic and cellulolytic microbial community in the rumen of dairy cows has been evolving rapidly. Using traditional culture-based methods, the main amylolytic and cellulolytic bacteria, fungi and protozoa in the rumen of dairy cows have been isolated. These culturable microbes have been found to only account for a small fraction of the total population of microorganisms present in the rumen. A more recent application of the culture-independent approach of metagenomics has acquired a more complete genetic structure and functional composition of the rumen microbial community. Metagenomics can be divided into functional metagenomics and sequencing-based computational metagenomics. Both approaches have been applied in determining the microbial composition and function in the rumen. With these approaches, novel microbial species as well as enzymes, especially glycosyl hydrolases, have been discovered. This review summarizes the current state of knowledge regarding the major amylolytic and cellulolytic microorganisms present in the rumen of dairy cows. The ruminal amylases and cellulases are briefly discussed. The application of metagenomics technology in investigating glycosyl hydrolases is provided and the novel enzymes are compared in terms of glycosyl hydrolase families related to amylolytic and cellulolytic activities.

## 1. Introduction

The rumen ecosystem harbours a vast number of microorganisms fermenting the ingested feedstuffs and producing various metabolites to meet the host’s nutritional requirement [1]. Nutritionists, microbiologists and physiologists, among others, have been studying the rumen microbial ecosystem in order to improve productivity and health and reduce the environmental impact of dairy cows.

Unlike ruminants in the wild, starch and cellulose are the principal components in diets for commercial dairy cows worldwide, providing the primary energy to the rumen microorganisms as well as the host [2]. Starch and cellulose degradation in the rumen have always been of key importance for dairy cows with numerous studies investigating the ruminal microbes and enzymes involved in starch and cellulose degradation [3,4]. Most of this research is based on more traditional approaches which include culturing and microscopy [4]. Over the last decades, increasingly more knowledge has been generated with the advancement of existing techniques and introduction of new analytical techniques.

The exploration of the species and enzyme activities involved in ruminal cellulose and starch digestion has been hampered by the limited number of rumen bacteria that can be cultured [5]. Metagenomics, a culture-independent analysis technique, has emerged in recent years as a powerful tool for exploring the collective structure and functioning of microbial genomes within a complex ecosystem. The application of metagenomics on rumen samples was first published in 2005 by Ferrer et al. [6] through functional screening technology. Since that, the metagenomic approach has been widely utilized to discover rumen microbial communities and enzymes. Li [7] discussed the periodic progress prior to 2015 of the metagenomics technologies in mining novel enzymes from the rumen microbiome, including fibrolytic and amylolytic enzymes. As high-throughput technologies have developed, sequence-based metagenomics combined with a functional metagenomic approach has been used, through which additional novel enzymes and metabolic activities were identified by comparison with multiple databases. The purpose of this review is to describe (1) our current understanding of the microbes and enzymes involved in starch and cellulose degradation in the rumen of dairy cows and (2) recent developments in technology where sequence-based and functional metagenomics can contribute to our knowledge of the structure and function of amylolytic and cellulolytic microorganisms in the rumen of dairy cows.

## 2. Starch Degradation in the Gastrointestinal Tract of Dairy Cows

Starch-rich grain is the primary energy component used in the modern diet for dairy cows, accounting for 20–40% of the ration of high-yielding cows. Due to the relatively high price of starch-containing ingredients, dietary starch should be used wisely to achieve cost-effectiveness and efficient production. Starch is a heterogeneous polysaccharide containing two structurally distinct α-linked polymers of glucose: amylose and amylopectin. The former is a linear D-glucose polymer containing ~99% α-1,4 links and the latter is the most abundant component of starch with 95% α-1,4 links and 5% α-1,6 links [8].

Unlike non-ruminants, starch degradation mainly occurs in the rumen, partly in the small intestine with the remainder fermented in the hindgut of ruminants. Starch degradation in each segment of the gastrointestinal tract is influenced by starch sources (e.g., corn, wheat, sorghum, barley) and processing (moistening, heating, or mechanical pressure) of the grain [4]. Data from 87 studies across a wide range of starch intakes (1–5.7 kg/d) showed that, on average, 71% of the starch intake was digested in the rumen [9]. Harmon et al. [10] analyzed data from 16 studies where the starch intake ranged from 1 to 5 kg/d and reported that ruminal starch digestion/fermentation was typically 75–80% of starch intake, with 35–60% of starch escaping rumen fermentation and digested in the small intestine. Between 35–50% of the starch that escapes small intestinal digestion was reported to be fermented in the large intestine. The starch digestion in the small intestine consists of three processes as reviewed previously [10]. Briefly, intestinal starch digestion starts in the lumen of the duodenum by the action of pancreatic α-amylase which hydrolyses amylose and amylopectin into maltose and other branched-chain products. The second process occurs at the brush border membrane via the action of the brush border carbohydrases (e.g., maltase, isomaltase) with the third process being glucose transportation from the intestinal lumen to the portal circulation [4].

### 2.1. Amylolytic Organisms in the Rumen

#### 2.1.1. Amylolytic Bacteria

The main starch-degrading microorganisms in the rumen are amylolytic bacteria, followed by protozoa and fungi [3]. Previous research has reported that bacterial digestion activities start with an attachment of bacteria to feed particles. The commonly reported amylolytic bacteria at present include *Streptococcus bovis*, *Ruminobacter amylophilus*, *Succinimonas amylolytica*, *Selenomonas ruminantium* and *Bifidobacterium* spp. (Table 1).

*Streptococcus bovis* can be easily isolated from the rumen fluid but only account for a small number of the total bacteria present in the rumen [11]. *Streptococcus bovis*, producing lactate as the main end-product, is present only when a large amount of starch or sugar is available as a substrate and the pH of the rumen fluid is low [12]. When conditions are favourable with high availability of starch or sugar, this species can grow explosively which leads to the overwhelming production of lactate and can result in rumen acidosis. *Ruminobacter amylophilus* is strictly anaerobic and Gram-negative with multiple shapes, arrangements and sizes. This species is capable of utilizing three forms of starch: amylose (linear α-1,4 linked glucose polymer), amylopectin (α-1,6 linkage) and pullulan (linear polymer of maltotriose residues linked by α-1,6 bonds) [13], mainly producing formate, acetate and succinate as end products. The starch molecules bind to cell surface receptors and are transported into the cell and hydrolyzed by intracellular amylase [14]. *Succinimonas amylolytica* is an anaerobic, Gram-negative and straight rod with rounded ends which can be motile with polar flagella. This species is less abundant among the ruminal bacteria when cattle are fed forage rations but is among the predominant bacteria when dietary starch is offered in the form of a grain mixture [15]. This species can hydrolyze starch producing succinate as the main product as well as a small amount of acetate and propionate. *Selenomonas ruminantium* is anaerobic and Gram-negative, and it consists of motile rods of 0.8–1.0 μm in width and 2–7 μm in length. This species was found to be more abundant in the rumen when animals were fed cereal grains compared to that fed roughage [16]. Most strains can ferment a wide range of substrates (Table 1). Lactate is the major fermentation end-product when high concentrations of glucose are present, but this is replaced by acetate and propionate at low glucose concentrations [12]. Besides the abovementioned amylolytic bacteria, some strains of the cellulolytic bacteria such as *Fibrobacter succinogenes*, *Butyrivibrio fibrisolvens* and *Clostridium* spp. are also capable of unitizing starch under certain conditions [17,18].

#### 2.1.2. Amylolytic Protozoa

The protozoa are also involved in degrading starch in the rumen. Between 20–45% of the amylolytic activities in the rumen have been attributed to protozoa [19]. The amylolytic protozoa digest starch through engulfment producing H_2_, CO_2_, acetate, butyrate and glycerol as products. However, the rate of uptake of starch grains varies greatly between species. The protozoa with high amylolytic activities include *Eremoplastron bovis*, *Diploplastron affine*, *Ophryoscolex caudatus* and *Polyplastron multiesiculatum*. The breakdown rate of starch by protozoa is by approximation determined by the initial starch or amylopectin concentration inside the protozoa [19]. Protozoa also have the capacity of slowing down the ruminal starch-fermentation rate because, on one the hand, protozoa ingest amylolytic bacteria resulting in a decrease in their populations [20] while on the other hand, they need at most 36 h to metabolize the engulfed starch granules [21].

#### 2.1.3. Amylolytic Fungi

Fungi account for a small proportion (~8%) of the rumen biomass where they are involved in degrading structural carbohydrates by producing a wide range of enzymes [22]. *Neocallimastix frontalis* was reported to hydrolyze starch by generating an endo-hydrolytic α-amylase from which maltose, maltotriose and maltotetraose were the major products [23]. Another three fungi species, *Orpinomyces joyonii*, *Neocallimastix patriciarum* and *Piromyces communis*, were also observed to be capable of digesting cereal grains [24].

### 2.2. Ruminal Starch-Degrading Enzymes

Due to their small size, bacteria cannot directly ingest starch granules or high-molecular-weight starch (e.g., amylopectin), but generate enzymes which specifically cleave the α-1,4 or α-1,6 bonds of amylose and amylopectin. These amylases can be typically classified into three main categories of hydrolytic activity: endoamylases, exoamylases and debranching enzymes (Table 2).

Endoamylases cleave the α-1,4 glucosidic linkages in the interior of the starch polymer or oligosaccharides in a random manner leading to the production of linear and branched oligosaccharides (Figure 1). α-Amylase is the most popular bacterial endoamylase which mainly hydrolyzes the internal α-1,4-bonds of amylose. A few types of α-amylases are also capable of hydrolyzing the α-1,6 bonds of amylopectin [13]. α-Amylases have been classified into the glycosyl hydrolases (GH) superfamily 13 and 57 based on amino acid sequence similarity [25]. Exoamylases hydrolyze the α-1,4 linkages at the nonreducing end of the starch molecule, of which the end-product is one predominant dextrin. β-Amylase which is classified into GH family 14 is an exoenzyme that liberates maltose by hydrolyzing 1,4-bonds. Because it cannot bypass 1,6-linkages, there always remain some β-limit dextrins after β-amylolysis. α-Glucosidases are members of GH family 15 and 31 which hydrolyze the α-1,4 or α-1,6 linkages on the nonreducing end in short saccharides produced by other enzymes. Glucoamylases have the ability to degrade both 1,4- and 1,6-linkages, solely forming glucose as an end product. Some debranching enzymes are also capable of cleaving the α-1,6 glucosyl link [26]. Isoamylases can degrade various branched structures of amylopectin, glycogen and branched oligosaccharides and dextrins. The pullulanase cleaves the α-1,6 link of pullulan-producing maltotriose which can then be hydrolyzed by isopullulanases yielding isopanose.

**Table 1 animals-12-03020-t001:** Fermentation characteristics of main ruminal amylolytic bacteria.

Microorganism	Substrate *	Fermentation Product	Gram Stain
*Streptococcus bovis*	Starch, maltose, cellobiose, sucrose, glucose, fructose, galactose, mannose, lactose (pectin, xylose, arabinose, mannitol, glycerol)	Lactate, CO_2_ (acetate, formate)	Positive
*Ruminobacter amylophilus*	Starch, maltose	Formate, acetate, succinate (lactate)	Negative
*Succinimonas amylolytica*	Starch, maltose, fructose	Succinate (acetate, propionate)	Negative
*Selenomonas ruminantium*	Maltose, cellobiose, xylose, arabinose, glucose, fructose, galactose, mannose, lactose, mannitol (starch, sucrose, glycerol, lactate)	Lactate, propionate, acetate, H_2_, CO_2_ (succinate)	Negative

Substrates or products in brackets indicate that they vary between strains. * From Hungate [11], Schaefer et al. [27] and Holdman et al. [28].

**Table 2 animals-12-03020-t002:** Enzymes involved in starch degradation in the rumen.

Category	Linkage	Enzyme	Substrate ^1^	End-Product ^2^
Endoamylase	Endo-α-1,4 glycosyl	α-amylase	Amylose, amylopectin, (granule, oligomer)	Linear and branched Oligosaccharides, glucose, maltose
Exoamylase	Exo-α-1,4 glycosyl	β-amylase	Amylose, amylopectin, oligomer (granule)	Maltose, β-limit-dextrin
	Exo-α-1,4 glycosyl	α-glucosidase	Oligomer (amylose, amylopectin)	Glucose
	Exo-α-1,4 glycosyl, exo-α-1,6 glycosyl	Glucoamylase	Amylose, amylopectin, oligomer (granule)	Glucose
Debranching enzyme	Endo-α-1,6 glycosyl	Isoamylase	Amylopectin	Linear oligosaccharide
	Endo-α-1,6 glycosyl	Pullulanase	Amylopectin	Maltotriose
	Endo-α-1,6 glycosyl	Isopullulanase	Amylopectin	Isopanose

The substrate in brackets indicates that fermentation depends on the enzyme source. ^1^ From Korarski, et al. [29], Robyt et al. [30], Fogarty [31], Vthinen [32]. ^2^ From Gomez et al. [33].

### 2.3. Factors Affecting Ruminal Starch Degradation

The rate and content of ruminal starch degradation vary with the type of cereal grains. Usually, wheat and barley starch are degraded more rapidly in the rumen than corn or sorghum starch [34]. Ruminal digestion of starch in the ground, rolled, or cracked corn (50–90%) or sorghum (42–89%) is generally lower than that in similarly processed barley (87–90%) [35]. Starch granules within the grain endosperm are surrounded by a protein matrix. The protein matrix in corn is extremely resistant to the invasion of amylolytic bacteria and can only be penetrated by some fungi, while for barley and wheat, the protein matrix is easily penetrated by a variety of proteolytic bacteria. In this regard, the combination of slowly and rapidly degraded grains was recommended [36].

Physical processing is another factor influencing ruminal starch degradation. Generally, processed grains are more digestible in the rumen [4]. With the rolling, cracking, or grinding of barley, a higher ratio of starch (87–90%) was digested in the rumen compared to the maize or sorghum (50–90%) [29]. Steam-flacking as a processing technology increased the grain starch degradation in the rumen, resulting in less starch available for the post ruminal fermentation [37].

Moreover, starch degradation in the rumen is influenced by intricate interrelations of multiple factors including diet composition, amount of feed consumed per unit time, mechanical alterations, chemical alterations and adapting degree of ruminal microbiota to the starch ratios in diet [3].

## 3. Cellulose Degradation

The rations for dairy cows are predominantly plant-based. The plant cell walls are primarily composed of cellulose which accounts for 20–30% of the dry weight of the primary cell wall. Cellulose is a homopolymer of glucose linked by linear 1,4-β-glycosidic bonds. Cellulose molecules associate with each other to form microfibrils in the form of crystalline formulations.

### 3.1. Cellulolytic Organisms in the Rumen

A large number of anaerobic bacteria, protozoa and fungi possess very efficient cellulolytic machinery which enables them to improve the feed conversion efficiency of cellulose. Cellulolytic organisms are those microbes predominantly digesting cellulose present in the diet, which were dominated mainly by bacteria, fungi and to a lesser extent the protozoa [38].

#### 3.1.1. Cellulolytic Bacteria

The *Ruminococcus flavefaciens*, *Ruminococcus albus* and *Fibrobacter succinogenes* are the major cellulolytic bacteria [38] (Table 3). *Fibrobacter succinogenes* is one of the most widespread cellulolytic bacteria in the rumen, which contributes 5–6% of the total prokaryotic 16S rRNA in the rumen contents of cattle [39]. The species is strictly anaerobic with Gram-negative cells. Their growth requires valerate and isobutyrate and partly need biotin and p-aminobenzoic acid [40]. *Fibrobacter succinogenes* strains have been reported to degrade cellulose, glucose and cellobiose mainly producing acetate and succinate [41]. Some strains are capable of degrading some cellulose allomorphs which are not susceptible to degradation by *Ruminococcus flavefaciens*. *Ruminococcus flavefaciens* are usually Gram-positive or Gram-variable and often generate a characteristic yellow pigment, particularly when grown on cellulose. Most *Ruminococcus flavefaciens* strains are able to degrade quite recalcitrant forms of cellulose which is difficult to digest by other species [42]. Previous research showed that *Ruminococcus flavefaciens* mainly attach to the cut edges of the epidermis, sclerenchyma and phloem cells when incubated with ryegrass leaves [43], and the attachment occurred at the epidermis and parenchyma bundle-sheath when incubated with orchard grass and Bermuda grass [44]. *Ruminococcus flavefaciens* mostly degrade cellulose and cellobiose, while some strains can also utilize glucose and other carbon compounds including maltose, lactose, xylose and starch. The main end products include acetate, succinate, formate and lactate, together with traces of hydrogen and CO_2_. *Ruminococcus albus* cells are usually single or diplococcic, 0.8–2.0 μm in diameter and Gram-negative to Gram-variable. Generally, in the rumen, *Ruminococcus albus* is more abundant than *Ruminococcus flavefaciens* [45]. *Ruminococcus albus* strains are able to degrade cellulose and cellobiose but cannot utilize glucose or other sugars. The main end-products of this degradation include acetate, ethanol, formate, lactate, hydrogen and CO_2_ with different combinations and proportions as the major products. *Ruminococcus albus* can produce ethanol, while the *Ruminococcus flavefaciens* produce succinate instead. The abovementioned three cellulolytic bacteria share some common features: (1) their growth needs a strict pH range from 6 to 7, (2) they are all strictly anaerobic and cannot survive when exposed to oxygen, (3) they digest cellulose by attachment to the cell surface through an extracellular glycocalyx and (4) these bacteria are majorly restricted to cellulose or the hydrolyzed products of cellulose.

Apart from the above three major bacteria, some strains in *Butyrivibrio fibrisolvens*, *Eubacterium cellulosolvens* and *Clostridium* spp. have also been reported to be involved indirectly in cellulolytic activities [38]. These cellulolytic bacteria degrade cellulose via adherence to an extracellular structure, the cellulosome. The processes for the adherence of bacteria to cellulose were reviewed by Miron et al. [49] and Krause et al. [38]. In short, the adherence could be defined in four steps: (1) non-motile bacteria are transported to the substrate, (2) bacteria adhere non-specifically to available sites on the plant cell wall, (3) the ligands or adhesins on the bacterial cell surface adhere specifically to the receptors on the substrate and (4) the adhered bacteria proliferate to create colonies on potentially digestible sites of a substrate.

#### 3.1.2. Cellulolytic Fungi and Protozoa

Cellulolytic activities have also been reported by fungal and protozoal populations in the rumen. The ruminal fungi with cellulolytic capacity include *Neocallimastix frontalis*, *Neocallimastix patriciarum* and *Neocallimastix joyonii*. The fungi also possess cellulosome-like machinery, which aids in the adherence process to cellulose [50]. Furthermore, cellulolytic protozoa, such as *Eudiplodinium maggie*, *Ostracodinium album* and *Epidinium caudatum* degrade cellulose by engulfment [51].

### 3.2. Ruminal Cellulose-Degrading Enzymes

Most cellulases are GH which are able to hydrolyze the glycosidic bonds within carbohydrate molecules [52]. In general, the hydrolases cleave the C-O, C-N or C-C bonds of the glucosides producing sugar and another compound, while cellulases mainly cleave the 1,4-β-glycosidic bonds between glucosyl moieties in cellulose into its monomers.

Cellulose is hydrolyzed to its monomeric glucose units by the synergistic action of three major types of cellulases: (1) endoglucanases (endo-1,4-β-D-glucan hydrolases), (2) exoglucanases (exo-1,4-β-D-glucan cellobiohydrolases) and (3) β-glucosidases (β-D-glucosidases) [38] (Table 4). These three cellulases break down cellulose at different sites and work synergistically on cellulose hydrolysis [53] (Figure 2). Briefly, the endoglucanase firstly randomly breaks down the amorphous regions of cellulose creating new chain ends, then the exoglucanases attack the non-reducing ends of cellulose or cellotetraose produced by endoglucanase, yielding cellobiose and cellotriose as products. The products are finally hydrolyzed to glucose by β-glucosidases.

All these abovementioned cellulases have been isolated from the ruminal cellulolytic microbes and classified into specific GH families, for example, the endoglucanases mainly belong to the GH family 5 and 9, whereas, exoglucanases are mostly present in the GH family 6, with the β-glucosidases mainly classified into GH family 3 [54].

## 4. Application of Metagenomics on Ruminal Microorganisms

To date, only a relatively small fraction of rumen microorganisms has been successfully isolated and cultured. Therefore, the largely unexplored microorganisms represent a significant untapped source of novel enzymes, especially those with multiple functions. Thanks to the development of next-generation sequencing technologies and bioinformatics tools together with the rapid progress in reference databases, metagenomics has become a powerful tool to study the rumen microbiome.

With metagenomics technologies, we can acquire the collective genetic structure and functional composition of rumen microorganisms without culturing their inhabitants. According to amino acid sequence similarity, GH and related enzymes are classified into specific families with all members in one family possessing the conserved catalytic mechanism. The public database of Carbohydrate Active enZyme (CAZy) describes the present knowledge on the enzyme families which build and breakdown complex carbohydrates and glycoconjugates. These families are classified based on experimentally characterized proteins and populated by sequences from public databases with significant similarity [55,56,57]. This database contains and updates all GH families that have been frequently used to mine enzymes in the rumen of dairy cows [57]. This section will summarize recent knowledge of the metagenomic insights into the starch- and cellulose-degrading enzymes in the rumen of dairy cows.

Rumen metagenomics analysis comprises two areas, including (1) functional metagenomics, in which the high-throughput screening technique is used for investigating gene products out of cloned expression libraries established by rumen metagenome DNA and (2) sequencing-based metagenomics in which the genomes and genes present in rumen microbes are explored through high-throughput next-generation sequencing.

### 4.1. Functional Metagenomics

Ferrer et al. [58] first applied the functional metagenomics approach in identifying hydrolytic enzymes involved in the ruminal digestion of plant polysaccharides, from which nine endoglucanases and twelve esterases were detected from the metagenomic library of dairy cows. Since then, more research has been conducted to investigate specific polysaccharide-degrading enzymes from the rumen through metagenomic libraries. Morgavi et al. [59] summarized the studies before 2012 about the applications of functional metagenomics for mining polysaccharide-degrading enzymes from the rumen (Table 5). In this review, the cellulose-degrading enzymes detected from cow rumen by those studies mainly belonged to GH families 5, 3 and 26. Li [7] reviewed the publications from 2012 to 2015, particularly on the lignocellulose-degrading enzymes mined from the rumen through functional metagenomic approaches (Table 5). They concluded that the new screened cellulases in the cow rumen mostly belonged to GH families 5, 8, 9 and 48. Even though the abovementioned studies have proven the applications of functional screening technique in characterizing ruminal enzymes, many challenges remain, e.g., (1) the expression libraries can only show a small fraction of functional diversity because not all target genes are easy to be expressed in foreign host systems and (2) the present techniques for detecting and screening desired functional activities need to be more efficient. To overcome these difficulties, new approaches have been developed. For instance, the habitat biasing methods were used to fractionate the microbial community in order to decrease the complexity of the microbiome or to enrich desired activities [60] or the combination of the in vitro compartmentalization and fluorescent-activated cell sorting was able to improve the functional screening of complex microbial ecosystems [6]. With further evolutions of techniques, new enzymes and metabolic activities will be characterized by the rumen microbiome with functional metagenomics.

### 4.2. Sequencing-Based Metagenomics

Sequencing-based metagenomics provides the collective genetic composition and functional activities of a microbial community at the DNA level. The first publication using next-generation sequencing-based rumen computational metagenomics for cataloguing the genes and activities involved in ruminal fibre degradation was reported in 2009 [57]. Later, Morgavi et al. [59] compared the contributions of four rumen fibrolytic bacteria to the GH families involved in plant polysaccharide degradation. The cellulose-degrading GH families 3, 5, 8, 9 and 51 were represented in the bacteria species of *Fibrobacter succinogenes* S85 and *Ruminococcus albus*. The publications since 2015 on metagenomics application related to ruminal starch- and cellulose-degrading enzymes of cows are summarized in Table 5. Most of these studies involved sequence-based metagenomics. Besides the cellulase GH families mentioned by Li [7], more novel cellulose-degrading enzymes were detected from rumen microbiomes and mostly belonged to GH families 44, 45, 6, 7, 88, 10, 51 and 95, while the starch-degrading enzymes were mainly from 13, 97, 31, 57, 77 and 15. Pitta et al. [69] characterized the rumen microbiome of dairy cow for functional pathways by lactation group (lactation 1, 2, 3 and 5) and stage of lactation (3 weeks before the anticipated freshening date, 1–5 days after the animal freshened, 4 weeks and 8 weeks into lactation) by a metagenomics approach, in which they found the predominance of GH13, GH27, GH77 and GH88 families that were actively involved with starch degradation. There were slight differences between lactation 1 and lactation 2 samples in the distribution of cellulases, endo-hemicellulases and debranching enzymes, while the oligosaccharide degrading enzymes were numerically higher in lactation 2 compared lactation 1 and 3. Gharechahi et al. [61] compared the fibre-attached rumen-uncultured microbiota and CAZyme produced after incubation with six lignocellulosic substrates, in which they found the most abundant GH families containing the GH3, GH31 and GH97 glucosidases and the GH51 endoglucanases. They also identified proteins that were the main components of cellulosome complexes but also had the potential to encode the α-amylases (GH13, GH13_6, GH13_7, GH13_15, GH13_28 and GH97) and cellulases (GH5, GH5_2, GH5_4, GH9, GH124 and GH128). Terry et al. [66] reviewed that the synergistic action of three classes of cellulolytic enzymes were involved in the breakdown of cellulose including endocellulase (GH family 5, 6, 7, 8, 9, 44, 45 and 48), exocellulase (GH 5, 6, 9 and 48) and β-glucosidase (GH 5 and 9). The literature shows that most metagenomics studies mainly focus on the ruminal fibrolytic activities and the efforts on starch degradation are relatively few.

In total, with the assistance of metagenomics tools, comprehensive studies as illustrated above will broaden our knowledge of the ruminal microbial structure and enzymatic activities, which in turn would allow for rumen manipulations to achieve a more efficient fibre and starch degradation. For instance, (1) as more microbial amylases and cellulases are identified out of the ruminal microbiome, it will be foreseeable to regulate the ruminal microbial amylolytic and cellulolytic activities through supplementing exogenous enzymes in the form of feed additive, (2) newly identified species will promote the process of isolating microbes out of the rumen and improve the development of microbe-culture techniques and (3) it will facilitate the commercial applications of rumen enzymes in various industries including feed additives and biofuel production.

## 5. Conclusions

This review has summarized the microbes and enzymes involved in starch and cellulose degradation and discussed the state of metagenomics technology in mining novel cellulase and amylase GH families in the rumen of dairy cows. To date, a number of amylolytic and cellulolytic microorganisms, their characteristics and their metabolic mechanisms in the rumen of dairy cows have been described. Still, uncharacterized microbes and enzymes need to be identified. The recently emerging technologies such as metagenomics have become more efficient in exploring new microbial species and strains, mining novel enzymes and monitoring microbial and enzymatic activities. This will improve the development of new culturing techniques. In turn, the advancement of our knowledge into the functioning of the microbiota of the rumen can facilitate the directed regulation of specific microbial activities or supplementation of exogenous enzymes.

## Figures and Tables

**Figure 1 animals-12-03020-f001:**
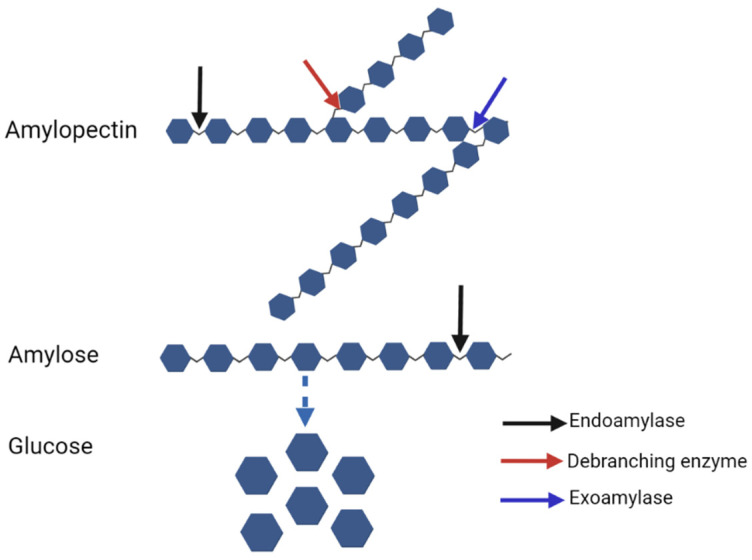
Starch breakdown by three types of amylases. Created in BioRender.com.

**Figure 2 animals-12-03020-f002:**
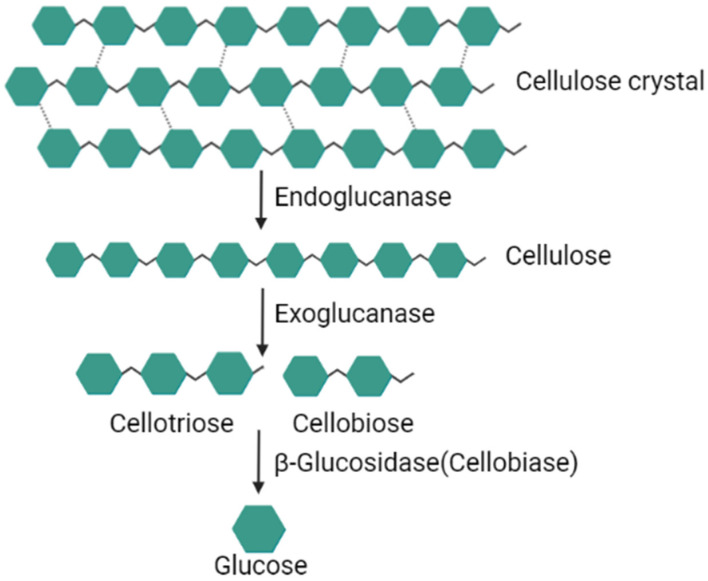
Structure cellulose breakdown by three cellulases. Created in BioRender.com.

**Table 3 animals-12-03020-t003:** Fermentation characteristics of main ruminal cellulolytic bacteria.

Microorganism	Substrate ^1^	Fermentation Product	Gram Stain
*Ruminococcus albus*	Cellulose, xylan, cellobiose (sucrose, xylose, arabinose, glucose, fructose, mannose, lactose, mannitol)	Acetate, ethanol H_2_, CO_2_ (formate, lactate)	Positive
*Ruminococcus flavefaciens*	Cellulose, xylan, cellobiose (sucrose, xylose, arabinose, glucose, mannose, lactose)	Acetate, succinate, H_2_, CO_2_ (formate, lactate)	Positive
*Fibrobacter succinogenes*	Cellulose, cellobiose, glucose (starch, pectin, maltose, lactose)	Acetate, succinate (formate, propionate, isovalerate)	Negative
*Butyrivibrio fibrisolvens*	Xylan, pectin, arabinose, glucose, fructose, galactose, mannose (starch, cellulose, maltose, cellobiose, sucrose, xylose, lactose)	Formate, butyrate, acetate, H_2_, CO_2_ (lactate, succinate)	Positive
*Clostridium polysaccharolyticum* ^2^	Starch, cellulose, xylan, pectin, maltose, cellobiose, xylose, arabinose (fructose)	Formate, butyrate, acetate, H_2_, (propionate)	Positive
*Clostridium longisporum* ^3^	Cellulose, pectin, cellobiose, sucrose, glucose, fructose, galactose, mannose, xylose, arabinose	Formate, butyrate, acetate	Positive
*Eubacterium cellulosolvens*	Cellulose, maltose, cellobiose, sucrose, glucose, fructose, lactose (xylan, pectin, galactose)	Lactate, H_2_ (formate, acetate, succinate, butyrate)	Positive

Substrates or products in brackets indicate that they vary between strains. ^1^ Substrate from Hungate [11], Holdman et al. [28], Schaefer et al. [27], van Gylswyk et al. [46]. ^2^ From van Gylswyk et al. [47], ^3^ From Varel [48].

**Table 4 animals-12-03020-t004:** Information on cellulose-degrading enzymes in the rumen.

Enzyme	Linkage	Substrate	Action
Endoglucanase	1,4-β-D-glucosidic linkage	Cellulose	cleave internal bonds at amorphous sites creating new chain ends
Exoglucanase	1,4-β-D-glucosidic linkage	Cellulose, cellotetraose	cleave two to four units from the non-reducing ends of the cellulose or cellotetraose molecules produced by endoglucanase
β-Glucosidase or cellobiase	1,4-β-D-glucosidic linkage	Cellobiose, cellotriose	hydrolyse the exoglucanase products into individual monosaccharides (glucose)

**Table 5 animals-12-03020-t005:** Amylolytic and cellulolytic enzymes mined from the rumen of dairy cows through metagenomics approach.

Reference	Enzyme	Glycoside Hydrolase Family
Gharechahi et al. [61]	Amylase	13, 97
	Cellulase	4, 5, 8, 9, 124, 128
	Endocellulase	74
Shen et al. [62]	Amylase	13, 15, 31, 57, 77
	Cellulase	97, 9, 5, 88, 45, 95, 44, 48
Zhao et al. [63]	Amylase	13, 15, 31, 4, 57, 63, 77, 97, 119
	Cellulase	5, 6, 8, 9, 10, 11, 26, 44, 45, 48
Wang et al. [64]	Amylase	13, 57
	Cellulase	5, 9, 88, 95
	Endocellulase	5, 6, 7, 9, 44, 45
	β-glucosidase	13, 88
Bohra et al. [65]	Cellulase	5, 9, 44, 45
Terry et al. [66]	Endocellulase	5, 6, 7, 8, 9, 44, 45, 48
	Exocellulase	5, 6, 9, 48
	β-glucosidase	5, 9
Jose et al. [67]	Cellulase	5
	Endocellulase	6, 7, 9, 44, 45
	β-glucosidase	1, 3
Shinkai et al. [68]	Cellulase	5, 6, 8, 9, 44, 45, 48, 74
Pitta et al. [69]	Amylase	13, 27, 77, 88
	Cellulase	5, 9, 48, 81
	Oligosaccharide degrading enzymes	1, 2, 3, 4, 13, 27, 29, 31, 35, 37, 38, 42, 57, 59, 63, 65, 88
Kang et al. [70]	Cellulase	5, 6, 7, 8, 9, 12, 44, 45, 48
Ko et al. [71]	Exocellulase	48
Gong et al. [72]	Endoglucanase cellulases	5, 8, 9
Hess et al. [73]	Unspecified	5, 8, 9, 10, 26
Zhao et al. [74]	α-amylase	57
Wang et al. [75]	β-glucosidase	3
	Endo-β-1,4-glucanase	5
Shedova et al. [76]	Endo-β-1,4-glucanase	5
Palackal et al. [77]	Glucanase/mannanase/xylanase	5, 26
Ferrer et al. [58]	Endo-glucanase	5, 26

## Data Availability

Not applicable.

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
