# Peer review of "Starch and Cellulose Degradation in the Rumen and Applications of Metagenomics on Ruminal Microorganisms"

_animals, 2022, doi:10.3390/ani12213020_

Round 1

Reviewer 1 Report

Mostly unknown microorganisms represent a huge source of new enzymes, especially those that perform many functions. Consequently, with the help of metagenomics tools and comprehensive studies will expand our knowledge about the structure of the microbial community of the rumen of dairy cattle as well as, the enzymatic activity of microorganisms, which, in turn, will allow manipulations with the rumen to achieve more effective degradation of fiber and starch.

The purpose of this review was to evaluate the current state of knowledge about amylolytic and cellulolytic microorganisms obtained by metagenomics present in the rumen of dairy cows and their role in the synthesis of enzymes and the establishment of end products of the fermentation process. 

The representatives of the microbiota of the rumen of cows – bacteria, protozoa and fungi, which are involved in the degradation of cellulose and starch, are considered. The features of the enzymatic activity of the microbiota in relation to feed carbohydrates and the final products of degradation in the rumen of cows are accessible. The application of metagenomics technology in the study of glycosyl hydrolases is presented and a comparison of new enzymes from the point of view of glycosyl hydrolase families related with amylolytic and cellulolytic activities is performed. In general, the work is relevant and of interest for further development. However, during the review process, some inaccuracies and suggestions to the authors were noted, to which we would like to receive a comprehensive answer:

1. In the first paragraph of the introduction there is no reference to the literature used (line 44-48).

2. Section 2 "Starch degradation in the rumen" is recommended to be replaced by "Starch degradation in the gastrointestinal tract of ruminants" (wish), because in some moments, after all, in this section, the authors cite the issue of splitting carbohydrates in the small and large intestines (Line 76, 92-100). 

3. In Table 2, you should make references to the literature used from which you received this information on each microorganism. If I realize correctly, the authors provide a link only to the original source of Stewart and Flint [24]. This correction would allow readers to more fully imagine what was directly done by the authors in this manuscript. 

4. Why did the authors not consider the amylolytic activity of Bifidobacterium sp. and Prevotella sp. in their review?

5. In Table 1, replace "species" with "sp.", or do the same. 

6. It is recommended to divide Table 1 into 2 and place each of them in the appropriate section (cellulolytic bacteria in the appropriate section of the manuscript), because in the section "2.1.1. Amylolytic bacteria" you considered only the amylolytic activity of bacteria.

7. References to the literature should also be given in table 2. To provide what exactly was done by the authors.

8. Line 107 was the number "sp." or the manifold "spp." used in Bifidobacterium?

Author Response

1. In the first paragraph of the introduction there is no reference to the literature used (line 44-48).

Reply: Thank you for the suggestion, two reference were cited.

2. Section 2 "Starch degradation in the rumen" is recommended to be replaced by "Starch degradation in the gastrointestinal tract of ruminants" (wish), because in some moments, after all, in this section, the authors cite the issue of splitting carbohydrates in the small and large intestines (Line 76, 92-100). 

Reply: Yes, I agree that this part is not only for the rumen degradation. I corrected the title according to your suggestion. Thank you so much.

3. In Table 2, you should make references to the literature used from which you received this information on each microorganism. If I realize correctly, the authors provide a link only to the original source of Stewart and Flint [24]. This correction would allow readers to more fully imagine what was directly done by the authors in this manuscript. 

Reply: Thank you so much for the comment. The more detailed reference information were added.

4. Why did the authors not consider the amylolytic activity of Bifidobacterium sp. and Prevotella sp. in their review?

Reply: Thank you for the nice suggestion. I have to clarify that this review is part of my project in which only the most commonly species were studied. So in this review I only elaborate the most frequently reported amylolytic bacteria which are the my desired bacteria.

5. In Table 1, replace "species" with "sp.", or do the same. 

Reply: Thank you for the correction. Already corrected.

6. It is recommended to divide Table 1 into 2 and place each of them in the appropriate section (cellulolytic bacteria in the appropriate section of the manuscript), because in the section "2.1.1. Amylolytic bacteria" you considered only the amylolytic activity of bacteria.

Reply: It is a super good suggestion. I separated the table to two parts and distributed to the related sections.

7. References to the literature should also be given in table 2. To provide what exactly was done by the authors.

Reply: Thank you for the suggestion. Most information in this table were originally cited from multiple literature. The detailed reference for this table was added.

8. Line 107 was the number "sp." or the manifold "spp." used in Bifidobacterium?

Reply: Thank you for the comments. The plural spp. is what I want to express. I already unified this phrase throughout the text.

Reviewer 2 Report

Dear authors,

 MS Animals 1947683, entitled: “Starch and cellulose degradation in the rumen and applications 2 of metagenomics on ruminal microorganisms”. This comprehensive review has merits and is of interesting for the scientific community, but there are some points that can be improved before acceptance for publication. They are listed below:

Line 17: Change to: “this review will summarized”

Line 42. Try to avoid words that are in the tittle, this will make you revision more findable

Line 71. The first part of the review is well summarized and match with the objective of the study, however, the second part of the review is short and some part needs to be better explored and more information is needed for the readers. For example, in table 4 authors listed several family numbers of Glycoside hydrolase, but I think that explaining more how these families are classified would help the readers to understand better what they means.

Line 100. I think a figure or scheme explaining how starch is degraded in the rumen, like the one used for cellulose would be interesting for the readers.

Table 1. Could be broken down in 2 tables separating amylolytic bacteria from cellulolytic bacteria. Also I suggest adding in the footnote a explanation for the brackets.

Lines 135-136. Authors stated that: “…Cellulolytic bacteria like…… are also capable of utilizing starch under certain conditions…” List the certain conditions with examples…

Line 215: I think in this section author should explore more the difference between the liquid and solid bacteria association on fiber degradation.

Line 346: Explain the difference among the families GH3, GH31 and GH97 and what are the implications of that.

Lines 350-352. I think author could give more details about the studies that are being cited to give a better understanding to the readers

References: As a review article I think that the references should be more updated, taking a looking on the references only 19% of the articles cited in the present review are in the range from 2012 to 2022.

Author Response

Line 17: Change to: “this review will summarized”

Thank you for this comment, I already corrected it.

Line 42. Try to avoid words that are in the tittle, this will make you revision more findable

Thank you for the suggestion.

Line 71. The first part of the review is well summarized and match with the objective of the study, however, the second part of the review is short and some part needs to be better explored and more information is needed for the readers. For example, in table 4 authors listed several family numbers of Glycoside hydrolase, but I think that explaining more how these families are classified would help the readers to understand better what they means.

Thank you for your nice suggestion.  I made a brief introduction about how the families are classified in line 275-279. The basic principle is based on the amino acid sequence similarity.

Line 100. I think a figure or scheme explaining how starch is degraded in the rumen, like the one used for cellulose would be interesting for the readers.

Thank you for this suggestion. I made a new figure that shows how the amylase catalyse the starch structure.

Table 1. Could be broken down in 2 tables separating amylolytic bacteria from cellulolytic bacteria. Also I suggest adding in the footnote a explanation for the brackets.

Yes, thank you so much for this nice suggestion. I divided the table into two parts.

Lines 135-136. Authors stated that: “…Cellulolytic bacteria like…… are also capable of utilizing starch under certain conditions…” List the certain conditions with examples…

Thank you so much for this suggestion. I listed the reference I cited for this statement. In this review, I would like to focus on the main digesters for starch and fiber. For the special conditions, I hope this references I cited here can help readers who need to get more information.

Line 215: I think in this section author should explore more the difference between the liquid and solid bacteria association on fiber degradation.

Thank you for your nice suggestion. For the first two sections, my originally thought for this review is to summarize the microbes and enzymes related to starch and fiber degradation in the rumen in general, not specifically to compare the difference of starch or fiber degradation in different phrases.

Line 346: Explain the difference among the families GH3, GH31 and GH97 and what are the implications of that.

Thank you for this comments. As I mentioned in the second paragraph of the third section. The GH families are classified based on the amino-acid sequence similarity or protein sequence and structure similarity, which is related to chemistry and microbiology. My background is animal nutrition, to clarify the differences between these families is really beyond my knowledge. I cited more reference about the CAZy databases in line 280, I hope if the readers are interested in the classification rules, these papers can help.

Lines 350-352. I think author could give more details about the studies that are being cited to give a better understanding to the readers

Thank you for the nice comment. Most papers I cited here is based on the current GH classification and is to evaluate some of the GH families. So they were not mining novel enzymes or having new findings. That was the reason I didn’t show too much details of them but to briefly show how many papers involved the carbohydrate active enzymes through the metagenomics approach are published. But I elaborated two more representative papers in line 324-332 and 337-340. I hope these could help the readers understand more.

References: As a review article I think that the references should be more updated, taking a looking on the references only 19% of the articles cited in the present review are in the range from 2012 to 2022.

Thank you for this nice comment. The reason is that most of my contents, e.g. the starch- and fiber-degrading microbes and the enzymes were most studied in last century. Even in the relatively new papers, most of their information were also indirectly cited from these references. Only the application of metagenomic is relatively new. I add several new papers as much as I can.

Reviewer 3 Report

Generally a well written and informative review covering starch and cellulose degradation and the microorganisms and enzymes involved in the rumen. There are just a few modifications that are suggested:

Simple summary.

Line 17; should read 'This review will summarize the'

Introduction.

Line 53; change 'degrading' to 'degradation'

Line 67-68; change to: 'As high-throughput technologies developed'

Line 73; change to: '2) recent developments in technology'

Starch degradation in the rumen

Line 123; It is not necessary to put in the words 'nonspore-forming' if the bacterium is Gram-negative.

Line 147; change 'population' to 'populations'. Each species of amylolytic bacteria is one population.

Cellulose degradation

Line 220; It is not necessary to indicate the bacterium is 'nonspore-forming' if it Gram-negative.

Application of metagenomics on ruminal microorganisms

Line 340; suggesting a change from 'Most of these studies were based on sequence-based metagenomics.' to 'Most of these studies involved sequence-based metagenomics.'

Conclusion

Line 369; change to 'cellulase and amylase GH families'

Author Response

Line 17; should read 'This review will summarize the'

Thank you for the correction. Already corrected in the text.

Introduction.

Line 53; change 'degrading' to 'degradation'

Thank you for the comments. Corrected.

Line 67-68; change to: 'As high-throughput technologies developed'

Thank you for the suggestion, I think your expression is more proper. Corrected.

Line 73; change to: '2) recent developments in technology'

Thank you for the suggestion. Corrected.

Starch degradation in the rumen

Line 123; It is not necessary to put in the words 'nonspore-forming' if the bacterium is Gram-negative.

Thank you for this knowledge. Corrected this throughout the text.

Line 147; change 'population' to 'populations'. Each species of amylolytic bacteria is one population.

Thank you for this knowledge. Changed.

Cellulose degradation

Line 220; It is not necessary to indicate the bacterium is 'nonspore-forming' if it Gram-negative.

Thank you for this knowledge. Corrected this throughout the text.

Application of metagenomics on ruminal microorganisms

Line 340; suggesting a change from 'Most of these studies were based on sequence-based metagenomics.' to 'Most of these studies involved sequence-based metagenomics.'

I agree with your suggestion. “Involve” is a more proper way.

Conclusion

Line 369; change to 'cellulase and amylase GH families'

Yes, changed.